# Balancing typological and dimensional approaches: Assessment of adult attachment styles with Factor Mixture Analysis

Fabia Morales-Vives[1,2]☉*, Gisela Ferré-Rey[1‡], Pere J. Ferrando[1,2☉], Misericòrdia Camps[1‡]

**1** Psychology Department, Universitat Rovira i Virgili, Tarragona, Spain, **2** Research Center for Behavior Assessment (CRAMC), Tarragona, Spain

☉ These authors contributed equally to this work.
‡ These authors also contributed equally to this work.
* fabia.morales@urv.cat

**Data Availability Statement:** The data is already available in the repository ZENODO: https://doi.org/10.5281/zenodo.4923526.

## Abstract

Many studies show the importance of adult attachment styles and their impact on social and emotional adaptation in adulthood. However, there is no agreement about whether attachment should be regarded as typological or dimensional, and some authors have proposed reconciling both options, so that adult attachment styles can be assessed more accurately and realistically. In this study we have adopted this comprehensive view and used Factor Mixture Analysis, the most appropriate model for assessing this mixture view. More specifically, we attempted to determine the nature and types (if any) of attachment styles that can be assessed with the Adult Attachment Questionnaire (CAA), using this mixture view. A total of 515 adults from Spain took part, with ages between 18 and 56 years old. In addition to the CAA questionnaire, they completed the Parental Bonding Instrument and the emotional stability subscale of the Overall Personality Assessment Scale. On the basis of the CAA scores, the results show that only two profiles–insecure attachment and normal-range–can be univocally differentiated. Furthermore, the results of a full multiple-group structural model show that each of these profiles has a different pattern of validity relations with the external variables maternal care, maternal overprotection and emotional stability. These differential validity results reinforce the general hypothesis that two differentiated clusters of individuals can be distinguished on the basis of the responses to the CAA items.

## Introduction

Initially proposed in the 1960s by John Bowlby, the attachment theory has given rise to a considerable number of studies that show the importance of attachment at different ages [e.g., 1, 2]. Bowlby [3] defined attachment as an enduring and strong affectionate bond to a particular person. This person, the attachment figure, becomes a source of security, support and reference, so the disruption or threats of disruption of this bond can cause anger, anxiety, depression or emotional detachment [3]. Ainsworth [4] identified three types of attachment in

**Funding:** Funded study: - Initials of authors who received any grant: FMV and PJFP - Grant numbers awarded to FMV and PJFP: PSI2017-82307-P and 2017 SGR 97 - Full name of funder: Spanish Ministry of Economy and Competitivity (PSI2017-82307-P) - Full name of funder: Catalan Ministry of Universities, Research and the Information Society (2017 SGR 97). URL of the Spanish Ministry of Economy and Competitivity: https://portal.mineco.gob.es/es-es/Paginas/default.aspx - URL of the Catalan Ministry of Universities, Research and the Information Society: https://agaur.gencat.cat/ca/inici - The funders had no role in study design, data collection and analysis, decision to publish, or preparation of the manuscript.

**Competing interests:** The authors have declared that no competing interests exist.

children: secure (when the caregivers are sensitive to their needs and respond in an affectionate manner), anxious/ambivalent (inconsistent caregiver responses that give rise to impulsive, attention seeking and helpless children), and avoidant (when caregivers are detached and do not respond to needs, children are characterized by lack of empathy, emotional detachment and antisocial behaviour). Although the attachment theory originally focused on the relationship between caregivers and children, several studies show that these bonds result in quite a stable pattern of interpersonal behaviours that in adulthood are known as adult attachment styles [5, 6].

Many studies show the importance of adult attachment styles, and their impact on social and emotional adaptation in adulthood. In fact, adult attachment styles are related to mental and physical health [7], emotional regulation [1], marital satisfaction [2], and the perceived support of family and friends and the search of social support in response to stress [8] among other variables. For this reason, it is important to have good instruments, with appropriate psychometric properties to properly assess attachment styles in adults.

Several self-report measures have been developed to assess adult attachment styles. One of the most important questionnaires in this field is the Hazan-Shaver attachment self-report [6], which focuses on adult romantic relationships. It assesses the following attachment styles, corresponding to three infant styles: a) secure, b) ambivalent, and c) avoidant. In fact, these authors considered that the three attachment patterns in childhood would emerge as three primary interpersonal styles in adolescence and adulthood [6]. However, Fraley & Waller [9] criticized traditional instruments such as the Hazan-Shaver attachment self-report for being categorical measures that assume that differences between people within a category are unimportant or inexistent (i.e. a typology). They considered this assumption to be unrealistic, and argued in favour of a change to a dimensional assessment of attachment styles. Although there is still no consensus on the typological or dimensional nature of attachment, Bartholomew & Horowitz [10] developed a model that reconciles both options. More specifically, they proposed a four-category model resulting from the combination of extreme positions in the dimensions of attachment anxiety (negative sense of self) and attachment avoidance (negative sense of others). According to this model, secure attachment involves a relative absence of attachment anxiety and attachment avoidance. Therefore, people with secure attachment have a positive image of themselves and they consider that they are worthy to be loved. Moreover, they trust others and they feel comfortable in their relationships. Preoccupied attachment involves high attachment anxiety and low attachment avoidance. These people have low self-esteem but a positive opinion of others. Although they desire intimacy with others, they do not trust that these other people will be available and supportive when needed. Dismissing attachment involves low attachment anxiety and high attachment avoidance. These people feel more comfortable in relationships with little intimacy and they do not expect support from others. Finally, fearful attachment involves high attachment anxiety and high attachment avoidance. These people have a negative opinion of themselves and do not trust others, they feel that they are not worthy of being loved and they are afraid of rejection. Preoccupied and fearful styles involve a high degree of dependency on close relationships, while secure and dismissing styles are linked to low degrees of dependency. Moreover, dismissing and fearful styles show a high degree of avoidance of intimacy in close relationships, unlike the secure and preoccupied styles. These authors developed the Relationship Questionnaire [10], which was also adapted for the Spanish population [11]. There is another version of this questionnaire, developed to detect the initial signs of potential future mother/infant relational problems, named Relationship Questionnaire-Clinical Version (RQ-CV) [12]. This version includes an additional attachment-style description labeled profoundly-distrustful attachment style. It refers to the distrust towards other people and the belief that everybody looks out for themselves, so nothing can be

expected from them. According to the authors [12], this description captures an attachment style that is not represented by the three standard insecure descriptions. RQ-CV is a simple instrument, easy to use, which also encompasses both categorical and dimensional perspectives.

Over the years various instruments have been proposed to assess adult attachment, with different psychometric properties and different underlying dimensions. This gave rise to frustration and confusion to the newcomers to the field. For this reason, Brennan, Clark, and Shaver [13] made an attempt at systemisation and developed a new measure to assess romantic attachment that would preserve the dimensions common to all the instruments. First of all, they administered all the questionnaires available to undergraduate students and factor-analysed the resulting scores. They obtained only two factors: a) anxiety (fear of abandonment and hypervigilance of romantic partners searching for signs of rejection), and b) avoidance (reluctance to get close to others). Next, they developed the questionnaire Experiences in Close Relationships (ECR) to assess these two factors. This questionnaire has been translated into many languages, including Spanish [14]. Fraley, Waller & Brennan [15] developed a revised version named Experiences in Close Relationships-Revised (ECR-R), which made changes to some of the items, especially in the subscale avoidance. However, according to Fraley [16], it should be taken into account that neither version assesses secure attachment with as much precision as insecure attachment.

The questionnaires mentioned above were adapted in Spain with samples of undergraduate students. For this reason, and taking into account the importance of having instruments that respect the particular characteristics of each country, in Spain Melero & Cantero [17] developed the Adult Attachment Questionnaire (CAA) with a heterogeneous sample of 445 adults. To do so, the authors reviewed the literature to identify all the variables that characterize each attachment style, and they included items about all these domains in the CAA. More specifically, they identified the following variables: self-concept, confidence in others, need for approval, dependency/autonomy/ self-sufficiency, consideration of relations as secondary, expression of feelings, discomfort with intimacy, conflict resolution strategies, lack of satisfaction with relationships, achievement orientation versus personal orientation, fear of relationships and interpersonal problems. They started with an initial set of 75 items, but the results of a principal component analysis with varimax rotation showed that 35 items functioned poorly. Therefore, they removed these items and carried out another principal component analysis with varimax rotation that yielded four components. The first component was named Low self-esteem, need of approval and fear of rejection (13 items), and it included items about dependency, worries about relationships, low self-esteem, fear of rejection, and behavioral and emotional inhibition problems. The second component was named Hostile resolution of conflicts, resentment and possessiveness (11 items), and it included items about the tendency to get angry, resentment, hostility and possessiveness. The third component was named Communication of feelings and comfort with relationships (9 items), and it included items about sociability, bilateral conflict resolution strategies, readiness to express feelings and confidence to explain problems to others. Finally, the fourth component was named Emotional self-sufficiency and discomfort with intimacy (7 items), and it included items about the need for individuality and autonomy rather than the establishment of affective bonds, and the avoidance of emotional commitment. Therefore, the first, second and fourth components refer to insecure attachment, while the third component refers to secure attachment.

In order to determine whether the CAA questionnaire allows subjects to be classified as having secure or insecure attachment, the authors carried out a k-means clustering with two clusters, using the participant's scores on each component. The first cluster contained all the participants with high scores on the first component, moderate/high scores on the second

component, low/moderate scores on the third component, and moderate/high scores on the fourth component. The second cluster contained those participants with low scores on the first component, low/moderate scores on the second component, moderate/high scores on the third component and low/moderate scores on the fourth component. Then, they carried out another k-means clustering with four clusters to find out if the participants could be classified in four groups, as Bartholomew & Horowitz [10] proposed. The first cluster was named fearful/hostile and it contained those participants with very high scores on the first and second components, low scores on the third component and moderate/high scores on the fourth. The second cluster was named preoccupied and contained the participants with high scores on the first component, moderate scores on the second and fourth component, and moderate/high scores on the third. The third cluster was named secure and contained the participants with very low scores on the first component, low scores on the second and fourth component, and high scores on the third. Finally, the fourth cluster was named distant, and contained the participants with low/moderate scores on the first and third components, moderate scores on the second component and low/moderate scores on the third. Considering these results, the authors proposed a system of norms that would allow the respondents to be classified according to their scores on the four components, with the scores being classified as very high, high, moderate/high, moderate, low/moderate, low, very low. Although this questionnaire has been widely used in Spain both by researchers and practitioners [18, 19], the norms do not classify many respondents because they have other combinations of scores. For this reason, the authors recommend a qualitative approach based on clinical judgment not only participants' scores. This approach, however, makes it difficult to use the instrument for research purposes because many studies involve large samples and the questionnaires are anonymous, which makes using clinical judgement unfeasible. For this reason, some authors have carried out k-means clustering with their own data, in order to classify their participants by using objective criteria [20]. Taking into account the limitations of the CAA questionnaire described, the current study focuses on this instrument, and uses a more appropriate methodological approach to determine the different profiles that the questionnaire can effectively differentiate.

At a general level, the present view regarding the nature of attachment measures can be regarded as a mixture or a hybrid view. On the one hand, these measures are assumed to have a dimensional structure but it is also thought that different profiles or clusters of individuals can be distinguished within this structure. We submit that this view is plausible but also that it cannot be completely assessed by the methodological approaches proposed to date. Thus, a factor analytic (FA) approach focuses solely on the dimensional structure of the measures, ignoring the possible clustering of individuals in different groups. And a pure Cluster analysis approach focuses only on distinguishing groups of individuals but ignores the dimensional structure of the data.

Conceptually, the mixed approach discussed above can be viewed as follows: the factor solution provides a k-dimensional basis on which individuals (or, to be more precise, their individual factor scores) can be represented as points. In this space, these points tend to cluster in different clouds, and the centroid of each cluster defines the profile of the corresponding group. Furthermore, our starting point is that the number and structure of the dimensions or factors of the CAA scores is known, and corresponds to the 4-dimensional solution described above. The number of Clusters or profiles, however, is less determinate, but we expect to find at least two profiles: a secure style and an insecure style. Taking into account this starting point, a combination between an exploratory and a confirmatory approach is required: the dimensional or FA part should be confirmatory, as we aim to confirm a structure previously proposed for the CAA items. This confirmation is the first aim of this study. The second aim of our study is determining the number of clusters of profiles that can be meaningfully

distinguished within the dimensional structure. The number of possible profiles must be determined in an exploratory fashion, as we can only stablish a plausible lower bound. We consider both aims to be of utmost importance if a more objective procedure for classifying all the potential respondents is to be obtained.

Finally, another aim of the current study is to determine whether the different profiles obtained through this mixture approach show the expected relationships with relevant external variables (i.e. evidence of validity, and also whether the patterns of relations are the same within the different profiles [i.e. differential validity]). More specifically, we expect to find that insecure attachment is related to lower emotional stability, lower perceived maternal care during childhood and higher perceived maternal overprotection during childhood, in comparison with more secure attachment profiles. Previous studies have already shown that insecure attachment involves lower levels of emotional stability [21, 22]. Furthermore, several studies have shown the importance of parental bonding, especially with the mother, during childhood for the development of attachment styles [23, 24]. In fact, greater maternal care and less maternal overprotection during childhood promote a more secure attachment style [23].

## Materials and methods

### Participants

This study involved the participation of 392 undergraduate students and 123 master's degree students (77% women) from Spain, with ages between 18 and 56 years old. The mean age and standard deviations were 23.2 and 6.3, respectively. Of the undergraduate students, 230 (58.7%) were student teachers, 59 (15.0%) were Social Education students, 46 (11.7%) were Social Work students and 57 were Psychology students (14.5%). The master's degree students were studying a master's degree in teacher training for compulsory secondary education and upper secondary education.

### Measures

**Adult Attachment Questionnaire (CAA) [17].**   This questionnaire assesses attachment styles in adults, as has been explained above. It consists of 40 items on a 6-point Likert scale (1 = Completely disagree, 6 = Completely agree). The principal component analysis with varimax rotation carried out by the authors [17] revealed the following four components: Low self-esteem, need of approval and fear of rejection (13 items, Cronbach's alpha = .86), Hostile resolution of conflicts, resentment and possessiveness (11 items, Cronbach's alpha = .80), Communication of feelings and comfort with relationships (9 items, Cronbach's alpha = .77), and Emotional self-sufficiency and discomfort with intimacy (7 items, Cronbach's alpha = .68).

**Parental Bonding Instrument (PBI) [25].**   We used the Spanish adaptation of this questionnaire [26]. This questionnaire assesses how individuals were treated by their parents during their first 16 years of life. Therefore, it is a retrospective measure, and can only be answered by subjects over 16. It has to be completed twice: once for the mother and once for the father, although in this study we only used the mother's version because in most cases the mother is the main attachment figure. It has 25 items that are answered on a 4-point scale, ranging from very unlikely to very likely, and it assesses the following dimensions: Care (12 items) and Overprotection (13 items). Care refers to perceived parental warmth and involvement contrasted with coldness and rejection, while Overprotection refers to perceived parental over-control contrasted with encouragement to autonomy. The internal consistency reliability estimates reported in the Spanish adaptation for each dimension were: α = .74 for Care and α = .82 for overprotection.

**Overall Personality Assessment Scale (OPERAS) [27].**   This questionnaire assesses the Big Five personality traits: Extraversion, Emotional stability, Conscientiousness, Agreeableness and Openness to experience. It has 40 items on a 5-point Likert scale (1 = Strongly disagree, 5 = Strongly agree). It provides scores free of the response biases social desirability and acqui-escence. These biases are corrected using the procedures proposed by Ferrando, Lorenzo-Seva and Chico [28] and Lorenzo-Seva and Ferrando [29]. In this study we only used the subscale of Emotional stability, which has an internal-consistency reliability estimate of .86.

## Procedure

The Ethics Commission for integrity in research, development and innovation (CEIR) of the X University (removed for peer review) approved this project. We also obtained a written informed consent from all participants, in accordance with the Declaration of Helsinki. This study was carried out in accordance with the recommendations of Spanish organic law 15/1999 and the Spanish Agency for Data Protection, which regulate the fundamental right to the protection of data. Questionnaires were administered collectively, in the classroom, by a trained psychologist. All the participants were guaranteed confidentiality, and assured that participation was voluntary. They were informed that the questionnaires were anonymous, so all the information they provided could not be traced back to them as individuals.

## Data analysis

A four-stage approach was used in the analyses. First, sample descriptive statistics were obtained in order to decide the most appropriate FA modeling (linear or non-linear) for the CAA item scores. Second, the proposed dimensionality and structure for the CAA were assessed by fitting a Confirmatory factor analytic (CFA) solution, in which the "cleanest" items that defined the 4-factor solution by Melero and Cantero [17] were specified as markers. The CFA model was fitted using Mplus v8.4 [30] with the following initial specifications: (a) eight items were excluded from the analyses (items 9, 11, 13, 15, 17, 33, 36, and 39) because a prelim-inary Exploratory factor analysis carried out in a pilot study with 100 students showed that they did not load on their corresponding factor, or had a small loading on their own factor, as it was also checked in the final sample; (b) in these preliminary analyses five complex items (5, 12, 18, 25, 26) loaded on a secondary factor, so they were also allowed to load on the expected secondary factor in the CFA.

In the third stage, provided that the structure obtained was clear, well-defined and in agree-ment with the hypothesized solution, a Factor Mixture Analysis (FMA) [31] was then carried out, again using Mplus. FMA can be seen as a hybrid between Latent class analysis (LCA) and FA, and is the most appropriate approach for assessing the mixture view discussed above [32]. More in detail, FMA assumes that (a) a common factor structure accounts for the CAA item responses (the structure tested in step 2), but also (b) that groups of individuals who behave characteristically (i.e. classes or profiles) can be identified in the factor space. Thus, the model assumes two sources of variation: dimensional, within-class variation, which reflects the com-mon structure of the data, and across-class variation, which models the differences between the groups or classes. In accordance with the rationale discussed above, we considered here the simplest FMA modelling, in which the CFA structure was assumed to be the same for all the classes (i.e. measurement invariance) but in which factor means (i.e. centroids) were allowed to vary in each class, so giving rise to the identifiable clusters discussed above.

In the fourth stage, evidence of validity was finally obtained via the relations among the dimensions and classes obtained in the previous stages and theoretically relevant measures. To obtain this evidence, the CFA model was extended to a full structural model in which the

scores on Care, Overprotection and Emotional stability were taken as external variables. The resulting standardized regression Beta weights obtained from fitting the model can be interpreted as structural coefficients corrected for measurement error.

## Results

### Descriptive statistics and CFA

The distributions of the CAA item scores were unimodal in all cases, and fairly symmetrical, with all the skewness values within the -1 to +1 range. Furthermore, the number of response points was rather high (6) and the sample not too large, which suggests that the linear FA model will be the best choice for this data [33]. The ordinal model in this case would have been expected to give rise to many sparse contingency tables and, therefore, potential instability.

The proposed CFA structure was fitted by using robust maximum-likelihood estimation (MLR) and the model-data fit was quite acceptable: (a) standardized root-mean-square residual (SRMS), 0.044; (b) root-mean-square error of approximation (RMSEA), 0.057, and (c) comparative fit index (CFI), 0.93. Measure (a) is an indicator of absolute fit, measure (b) indicates relative fit, and (c) is a measure of comparative fit with respect to the null independence model. Thus, in all three facets, the fit can be considered acceptable. Table 1 shows the resulting pattern matrix. All the items have substantial loadings on the expected factors, which makes interpretation quite clear. The first factor corresponds to the factor that Melero and Cantero [17] named Low self-esteem, need for approval and fear of rejection, the second factor corresponds to Hostile resolution of conflicts, resentment and possessiveness, the third factor corresponds to Communication of feelings and comfort with relationships and the fourth factor corresponds to Emotional self-sufficiency and discomfort with intimacy. Table 2 shows the inter-factor correlation matrix. As can be seen, factor three has negative correlations with the other factors (although the correlation with factor 2 was nonsignificant). This was expected because this factor is related to positive attachment, unlike the other factors, which are more related to attachment problems.

### FMAs

The CFA model described above, as fitted with MLR, can also be interpreted as an FMA in which a single class is specified (i.e. the baseline FMA model). The acceptable fit found for this model gives strong support for the assumption of a dimensional structure underlying the CAA scores. What remains to be done now is to assess whether specifying more than a single class improves the relative fit of the model and, if it does, how many classes provide the best relative fit. Therefore, a sequence of FMA models specifying a range of two to four classes was fitted to the data by using MLR estimation. Relative fit was assessed using three groups of indicators [34]. The first indicator was a parsimony information criterion (in this case the BIC), which provides a trade-off between simplicity and goodness-of-fit. The second was the normed entropy criterion, which provides values between 0 and 1, and indicates the extent to which individuals can be differentiated in terms of the class they belong to, or, in other words, the ability of the solution to provide well separated classes. And the third was a difference test, in our case the Lo-Mendel-Rubin (LMR) test, which assesses whether adding a new class to the previous number of classes significantly improves the fit of the model. Results are shown in Table 3, and they can be summarized as follows. All the BIC values for the solutions with 2 or more classes are better than the single-class value (i.e. the standard CFA solution). Second, the BIC values for 2, 3, and 4 classes are very similar (relative differences are about 0.08%) and do not allow a clear choice to be made. Third, all entropy values are above the conventional proposed cut-off value of 0.80 [30], which also makes it hard to take a clear decision. However, the

**Table 1. Pattern matrix obtained in the confirmatory factor analysis.**

| Item | F1 | F2 | F3 | F4 |
|---|---|---|---|---|
| 3. Con frecuencia, a pesar de estar con gente importante para mí, me siento sólo/a y falto de cariño (*Even if I am with people who are important to me, I often feel alone and unloved*) | **.55** | .00 | .00 | .00 |
| 8. No suelo estar a la altura de los demás (*I often feel I am not in the same league as other people*) | **.70** | .00 | .00 | .00 |
| 10. I like having a partner, but I am afraid of being rejected (*Me gusta tener pareja, pero temo ser rechazado/a por ella*) | **.50** | .00 | .00 | .00 |
| 12. When I have a problem with someone else, I cannot stop thinking about it (*Cuando tengo un problema con otra persona, no puedo dejar de pensar en ello*) | **.56** | .00 | .42 | .00 |
| 14. I have feelings of inferiority (*Tengo sentimientos de inferioridad*) | **.77** | .00 | .00 | .00 |
| 18. I am very sensitive to the criticism of others (*Soy muy sensible a las críticas de los demás*) | **.77** | .00 | .45 | .00 |
| 21. I have confidence in myself (*Tengo confianza en mí mismo*) | **.73** | .00 | .00 | .00 |
| 23. I find it difficult to take a decision unless I know what other people think (*Me resulta difícil tomar una decisión a menos que sepa lo que piensan los demás*) | **.45** | .00 | .00 | .00 |
| 26. I worry a lot about what people think of me (*Me preocupa mucho lo que la gente piensa de mí*) | **.81** | .00 | .37 | .00 |
| 30. I would like to change a lot of things about myself (*Me gustaría cambiar muchas cosas de mí mismo*) | **.66** | .00 | .00 | .00 |
| 34. I feel I need more care than most people (*Siento que necesito más cuidados que la mayoría de las personas*) | **.44** | .00 | .00 | .00 |
| 37. I find it difficult to break off a relationship because I am afraid I won't be able to cope (*Me cuesta romper una relación por temor a no saber afrontarlo*) | **.45** | .00 | .00 | .00 |
| 2. I admit no discussion if I think I am right (*No admito discusiones si creo que tengo razón*) | .00 | **.51** | .00 | .00 |
| 4. I believe in "an eye for an eye and a tooth for a tooth" (*Soy partidario/a del "ojo por ojo y diente por diente"*) | .00 | **.62** | .00 | .00 |
| 7. If a member of my family or a friend contradicts me, I easily get angry (*Si alguien de mi familia o un amigo/a me lleva la contraria, me enfado con facilidad*) | .00 | **.66** | .00 | .00 |
| 20. When there is a difference of opinion, I insist that my point of view is accepted (*Cuando existe una diferencia de opiniones, insisto mucho para que se acepte mi punto de vista*) | .00 | **.47** | .00 | .00 |
| 24. I am resentful (*Soy rencoroso*) | .00 | **.68** | .00 | .00 |
| 29. When I get angry with someone else, I try to make sure that it is they who come to apologise (*Cuando me enfado con otra persona, intento conseguir que sea ella la que venga a disculparse*) | .00 | **.64** | .00 | .00 |
| 31. If I had a partner who told me that they found someone of the opposite sex attractive, I would be very upset (*Si tuviera pareja y me comentara que alguien del sexo contrario le parece atractivo, me molestaría mucho*) | .00 | **.33** | .00 | .00 |
| 1. I find it easy to express my feelings and emotions (*Tengo facilidad para expresar mis sentimientos y emociones*) | .00 | .00 | **.59** | .00 |
| 5. I need to share my feelings (*Necesito compartir mis sentimientos*) | .34 | .00 | **.76** | .00 |
| 16. I feel comfortable at parties and social gatherings (*Me siento cómodo/a en las fiestas o reuniones sociales*) | .00 | .00 | **.60** | .00 |
| 27. When I have a problem with someone else, I try to talk it over with them and find a solution (*Cuando tengo un problema con otra persona, intento hablar con ella para resolverlo*) | .00 | .00 | **.41** | .00 |
| 32. When I have a problem, I tell someone else who I trust (*Cuando tengo un problema, se lo cuento a una persona con la que tengo confianza*) | .00 | .00 | **.55** | .00 |
| 35. I prefer being by myself to being sociable (*Soy una persona que prefiere la soledad a las relaciones sociales*) | .00 | .00 | **.58** | .00 |
| 38. Other people think that I am an open person who is easy to get to know (*Los demás opinan que soy una persona abierta y fácil de conocer*) | .00 | .00 | **.57** | .00 |
| 40. I see that people often trust me and value my opinions (*Noto que la gente suele confiar en mí y que valoran mis opiniones*) | .00 | .00 | **.48** | .00 |
| 6. I never manage to commit seriously to the relationships I have (*Nunca llego a comprometerme seriamente en mis relaciones*) | .00 | .00 | .00 | **.67** |

(*Continued*)

**Table 1.** (Continued)

| Item | F1 | F2 | F3 | F4 |
|---|---|---|---|---|
| 19. When someone becomes dependent on me, I need to distance myself (*Cuando alguien se muestra dependiente de mí, necesito distanciarme*) | .00 | .00 | .00 | **.35** |
| 22. (*No mantendría relaciones de pareja estables para no perder mi autonomía*) | .00 | .00 | .00 | **.60** |
| 25. I prefer stable relationships to sporadic partners (*Prefiero relaciones estables a parejas esporádicas*) | -.23 | .00 | .00 | **.39** |
| 28. I like having a partner but, at the same time, I find it smothering (*Me gusta tener pareja, pero al mismo tiempo me agobia*) | .00 | .00 | .00 | **.66** |

*Note*. F1 = Low self-esteem, need of approval and fear of rejection; F2 = Hostile resolution of conflicts, resentment and possessiveness; F3 = Communication of feelings and comfort with relationships; F4 = Emotional self-sufficiency and discomfort with intimacy.

LMR results seem quite clear: there is a significant improvement between one and two classes, but increasing the number of classes above two does not significantly improve the model-data fit. So, the simpler two-class FMA solution seems to be the best choice for this data.

Inspection of the 2-class solution revealed the characteristics of each profile. More specifically, there is a general profile, which has been named normal-range, which encompasses 77.48% of participants, and a differentiated profile, which has been named insecure attachment, which encompasses 22.52%. Table 4 shows the mean factor scores for the insecure profile (for identification purposes, the means for the secure attachment have been set to zero). As can be seen in Table 4, the differentiated profile has high means for factors 1, 2 and especially 4, and a low mean for factor 3. All of these means differ significantly from zero, which means that all of them differ from those in the general group. The effect sizes are medium for factors 1 and 3, small for factor 2 and very large for factor 4. Therefore, the two groups differ above all in terms of factor 4. Overall, in comparison with the undifferentiated profile, the insecure attachment group shows high levels of fear of rejection, need of approval and low self-esteem, high levels of hostile resolution of conflicts, resentment and possessiveness, very high levels of emotional self-sufficiency and discomfort with intimacy, and low levels of communication of feelings and comfort with relationships. It is because of these results that this class has been named insecure attachment. The other group has been named normal-range because it includes a large number of subjects in which no further well-defined classes can be differentiated. However, in comparison with the insecure attachment group, this class has, in general, a more secure and positive attachment.

**Table 2.** Inter-factor correlations.

| CESF | F1 | F2 | F3 | F4 |
|---|---|---|---|---|
| F1 | - | | | |
| F2 | .40** | - | | |
| F3 | -.50** | -.09 | - | |
| F4 | .21** | .23** | -.23** | - |

*Note*. F1 = Low self-esteem, need of approval and fear of rejection; F2 = Hostile resolution of conflicts, resentment and possessiveness; F3 = Communication of feelings and comfort with relationships; F4 = Emotional self-sufficiency and discomfort with intimacy.

** $p < .01$

**Table 3. Fit indices for the mixture analyses based on the 4-factor CAA model.**

| N° of classes | BIC | Δ | LMR test | p |
|---|---|---|---|---|
| 1 | 55267.22 | - | - | - |
| 2 | 55072.76 | .95 | 225.67 | .00 |
| 3 | 55025.77 | .96 | 303.75 | .24 |
| 4 | 55045.18 | .93 | 315.65 | .21 |

*Note.* Criteria: Bayesian Information Criterion (BIC), Entropy value (Δ), Lo-Mendel-Rubin (LMR) difference test with associated probability.

## Validity assessment: Extended CFA model

Evidence of validity based on theoretically-related external variables was assessed by using a full multiple-group structural model as follows. The measurement part of the model was the two-group extension of the 4-factor CFA model in the previous section, in which (a) the measurement parameters were assumed to be invariant in both groups, and (b) the groups were the two classes (insecure attachment and normal-range) identified in the FMA solution above. So, the measurement part of the model is, essentially, the final FMA model that we considered as the most appropriate for the CAA scores in the previous section.

In the structural part of the model, the scores on Care, Overprotection and Emotional stability were taken as external variables theoretically related to the four CAA factors. The status of these variables as indicators in the model, however, differs. While Emotional stability can be plausibly treated simply as a variable that can be predicted from the CAA factors (as in conventional validity assessment), Care and Overprotection are clearly explanatory variables for the CAA factors. More specifically, as Care and Overprotection refer to the mother's behavior towards the son or daughter, these variables must be considered as precursors of the adult attachment style developed later. So, they must be treated as formative rather than effect indicators [35]. These specifications imply that, in the case of Care and Overprotection, the latent CAA factors are taken as dependent variables to be predicted (or explained) by Care and Overprotection, whereas, in the case of emotional stability, this indicator is simply taken as a dependent external variable that can be predicted from the latent CAA factors.

Given the exploratory nature of the validity assessment, the regressions concerned with the external variables were not constrained to be the same in each group. So, the measurement part of the model is group-invariant (which provides a strong basis for model identification), but the structural part is not. Finally, because in the validity part of the model only single indicators (test scores) were used as external variables, the standardized regression weights were corrected for the measurement error of the single indicators by using the corresponding

**Table 4. Mean factor levels in each class according to the Mixture Factor Model.**

| Latent classes | F1 | F2 | F3 | F4 |
|---|---|---|---|---|
| Insecure attachment | 0.45** | 0.29** | -0.36** | 3.19** |
| | d = 0.54 | d = 0.39 | d = 0.40 | d = 5.22 |
| Secure attachment | 0.00 | 0.00 | 0.00 | 0.00 |
| | (fixed) | (fixed) | (fixed) | (fixed) |

*Note.* d = Cohen's effect size; F1 = Low self-esteem, need of approval and fear of rejection; F2 = Hostile resolution of conflicts, resentment and possessiveness; F3 = Communication of feelings and comfort with relationships; F4 = Emotional self-sufficiency and discomfort with intimacy.

** p < .01

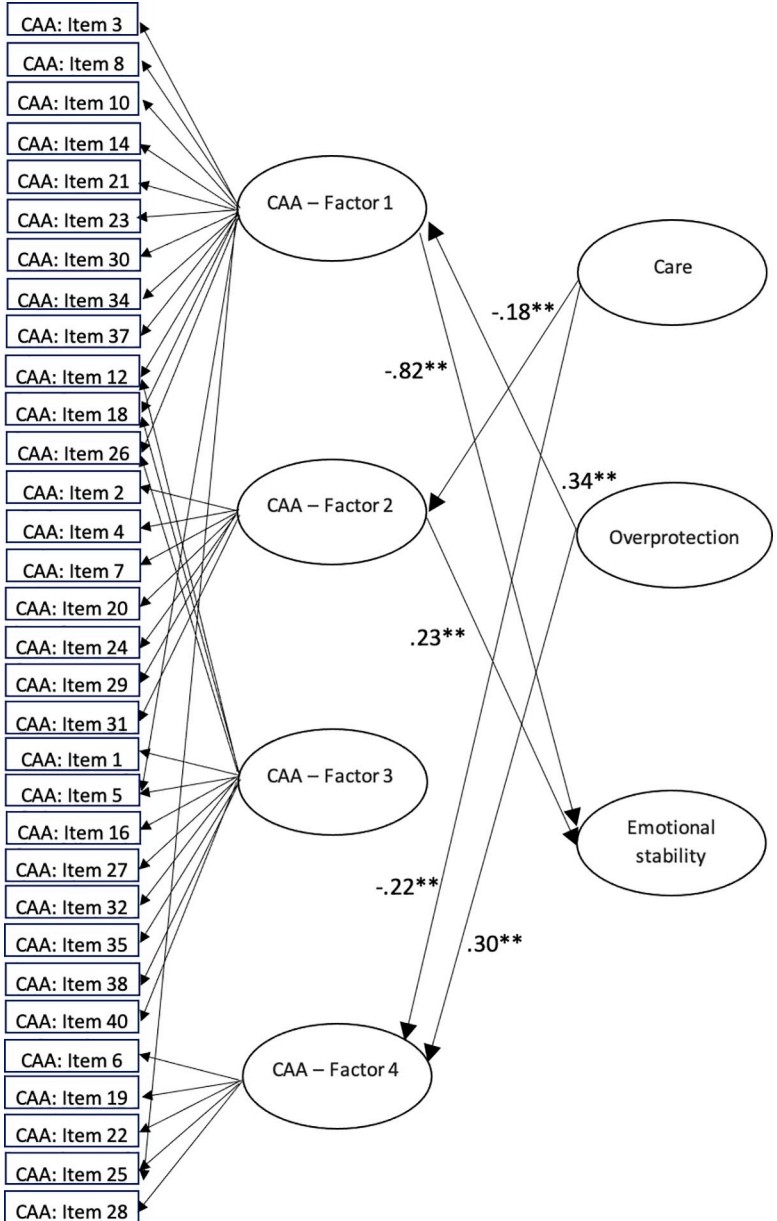

**Fig 1. Structural equation model for the insecure attachment group.**

reliability estimates [36]. The model summarized so far was fitted using the same procedures as in the CFA above, and the model-data fit was quite acceptable: SRMS = 0.07; RMSEA = 0.041, and CFI = 0.93. The standardized regression coefficients for each group can be seen in Figs 1 and 2. Because of the intersection of so many lines, in order not to make these two figures even more difficult to understand, neither the residual paths nor the inter-factor correlations have been represented (inter-factor correlations are shown in Table 2). Moreover, the multiple correlation estimate between Emotional stability and the four CAA factors was .75 in the insecure attachment group and .80 in the normal-range group.

Apart from the structural relations with the external variables (Care, Overprotection and Emotional stability), it is also of interest to assess whether the average levels of these variables

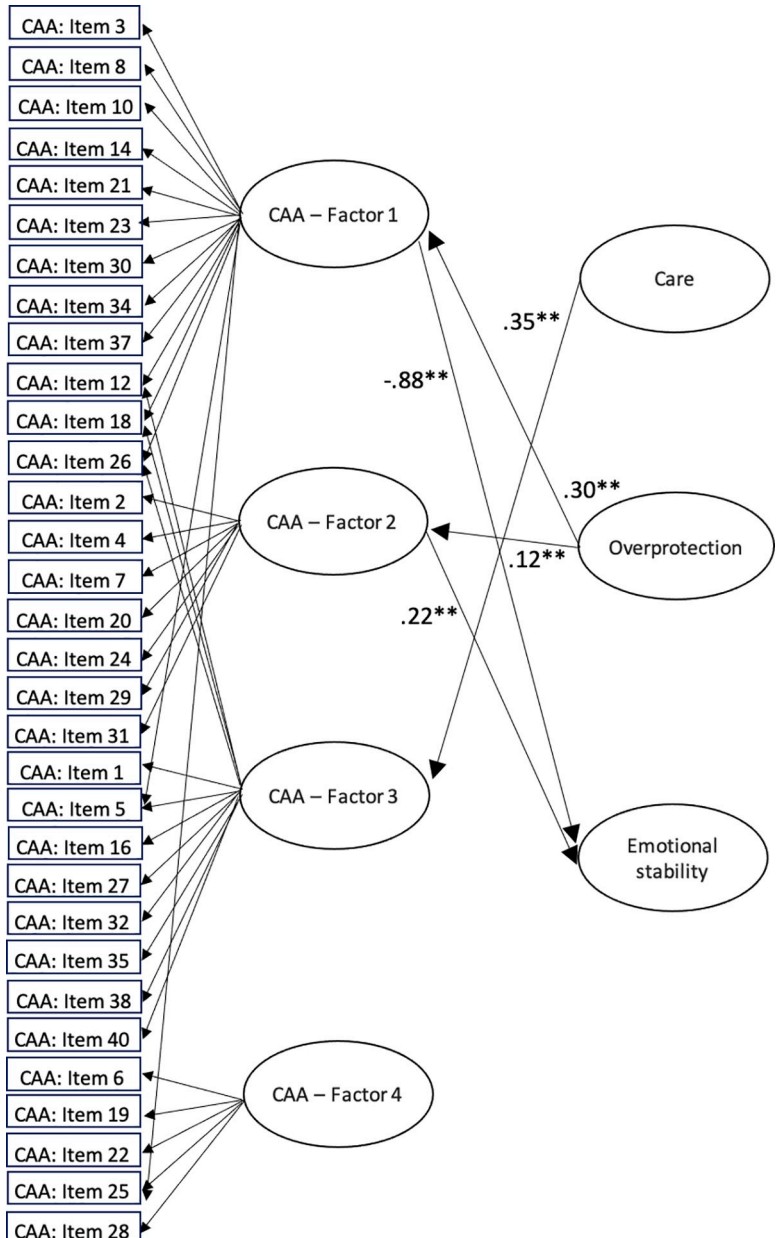

**Fig 2. Structural equation model for the normal-range attachment group.**

differ between the two groups, and this is the final piece of information derived from the structural model. As in the FMA analysis, the means on the external variables were set to zero in the largest, undifferentiated group, and freely estimated in the insecure attachment group, here with the restriction that the pooled variance in both groups is 1. The results are in Table 5 and are all significant at the conventional .05 level (standard errors are in the order of 0.04 in all cases). Furthermore, given the scaling we used, the mean estimates in the insecure attachment group are also Cohen's *d* effect sizes for the difference of means. Under the usual standards for *d*, they would qualify as small or medium. According to the results, the insecure attachment group has lower scores for Emotional stability and Care, and higher scores for Overprotection than the normal-range group.

**Table 5. Means for each class in care, overprotection and emotional stability.**

| Latent classes | Care | Overprotection | Emotional stability |
|---|---|---|---|
| Insecure attachment | -0.23** | 0.26** | -0.30** |
| Secure attachment | 0.00 | 0.00 | 0.00 |
| | (fixed) | (fixed) | (fixed) |

** $p < .01$

## Discussion

Adult attachment styles have traditionally been regarded as typological, so for many years instruments were developed with this assumption as a starting point. However, authors such as Fraley & Waller [9] criticized this approach for being unrealistic. Indeed, under this approach, the inter-item relations between measures of adult attachment are solely accounted for by class membership, and are inexistent within each class. According to this quite different perspective, attachment styles should be regarded as dimensional instead of categorical. Finally, Bartholomew & Horowitz [10] propose reconciling both options in order to be able to assess adult attachment styles more accurately and realistically. In this study we have adopted this comprehensive view, and have aimed at determining which attachment styles can be assessed with the CAA questionnaire by using a methodological approach that is more appropriate than the one used so far. More specifically, we have used the Factor Mixture Analysis model, a hybrid between Latent Class Analysis (LCA) and Factor Analysis that allows the structure derived from the item responses to be simultaneously both categorical and dimensional. This analysis uses continuous latent variables so that individuals can have different levels within each group. Therefore, it reconciles the traditional typological view with the dimensional view of attachment styles.

The first step in our mixed approach was to confirm the four factor structure proposed by Melero and Cantero [17]. The results show that the model-data fit is quite acceptable, and the resulting structure is clear and strong. As expected, there is a first factor related to low self-esteem, need of approval and fear of rejection, a second factor related to hostile resolution of conflicts, resentment and possessiveness, a third factor related to communication of feelings and comfort with relationships, and a fourth factor related to emotional self-sufficiency and discomfort with intimacy. The first, second and fourth factors refer to insecure attachment, while the third component refers to secure attachment. These results support that there is a dimensional structure underlying the CAA scores, as was expected.

The second step was to determine if specifying more than one profile (or latent class) improves the relative fit of the model and, if it does, how many profiles provide the best relative fit. The results show that there is a significant improvement when two profiles are specified, in comparison with the single profile solution, but that a further increase in the number of profiles does not significantly improve the model-data fit. Therefore, we chose the solution with only two profiles. In fact, we expected to find at least two profiles: one for insecure attachment and one for secure attachment. However, what we found was a general profile (77.48% of participants) and a specific profile. Taking into account the means obtained by each group in the factors related to attachment, explained above, we called the first profile as normal-range and the second profile as insecure attachment. These two profiles differ above all in the fourth factor, since the insecure attachment group scored much higher than the other group on this factor. Therefore, individuals with insecure attachment are characterized in particular by their high levels of emotional self-sufficiency and discomfort with intimacy, and it is this aspect that

is largely responsible for the two different profiles. According to Bartholomew & Horowitz [10], this feature is characteristic of dismissive/avoidant insecure attachment. This attachment style involves denying or minimizing the importance of attachment relationships as a defence against rejection, or for fear of losing the people with whom they have formed an affective bond. For this reason, the feeling of self-sufficiency and independence is important to them.

The results of the full multiple-group structural model show that there is a different pattern of relationships for each profile between the four factors of CAA questionnaire and the external variables Care, Overprotection and Emotional stability. The fact that the structural relations (particularly those based on the formative indicators) differ among the two groups reinforces the general hypothesis that two differentiated clusters of individuals can be distinguished on the basis of their responses to the CAA items. Thus, the internal FMA analysis suggests that two different profiles emerge from these responses, and the results of the extended validity model suggest that these profiles also relate differently to relevant external variables. More specifically, the results suggest that lower levels of perceived maternal care in the group with insecure attachment are related to increased problems in some attachment-related factors as these participants show higher levels of hostile resolution of conflicts, resentment and possessiveness, and also increased emotional self-sufficiency and discomfort with intimacy. Furthermore, higher levels of perceived maternal overprotection in this group are related to a greater fear of rejection, need of approval and low self-esteem, and also greater emotional self-sufficiency and discomfort with intimacy. These results are congruent with the study by Wilhelm, Gillis & Parker [24], which shows that maternal bonding during childhood has an effect on the development of adult attachment. However, according to this previous study, gender is a moderating factor in the relationship between responses to parenting style and adult attachment. The current study is not focused on gender differences in validity relations, but further studies should be done in order to know if these differences between men and women are replicated using the instruments and the methodology of the current study. Moreover, Páez et al. [23] showed the negative effect of less maternal care or greater maternal overprotection on the development of adult attachment. However, maternal care and overprotection are not related to the communication of feelings and comfort with relationships in this group, while they are in the group with a more secure attachment style. In fact, the maternal care perceived by the normal-range group is related to higher levels of communication of feelings and comfort with relationships. But perceived maternal care has no relationship with the other three CAA factors in this group (the factors that involve attachment problems). Higher perceived maternal overprotection also has negative consequences for this group, as it is related with increased fear of rejection, need of approval and low self-esteem, and increased hostile resolution of conflicts, resentment and possessiveness.

Overall, the differential validity results suggest that perceived maternal care and overprotection are related to different factors of adult attachment depending on the group. However, the pattern of relationships with emotional stability is the same for both groups. Several studies have shown that insecure attachment involves lower levels of emotional stability [21, 22], which is congruent with the negative relationship found between this personality trait and the factor related to low self-esteem, need of approval and fear of rejection. However, there is a significant positive relationship between emotional stability and the factor related to hostile resolution of conflicts, resentment and possessiveness. This was unexpected since previous studies have shown a negative relationship between emotional stability and hostility [37, 38]. Therefore, further evidence of generalizability is needed, as no previous studies on this issue were based on the CAA questionnaire. The result may be explained by the fact that the factors in this questionnaire assess a broad construct that involves several facets. In fact, Moon, Hollenbeck, Marinova and Humphrey [39] showed that a broad factor is likely to contain items that

vary in terms of their theoretical link to an external variable. Some of these items will be positively related to this variable, and other items negatively related, which makes it difficult to predict and interpret the correlation between the overall measure and the external variable. But, in any case, further studies should be done to determine whether the result replicates in new samples, and, if it does, what the reason is for this positive relationship.

The group means for care, overprotection and emotional stability follow the theoretically expected pattern. The insecure attachment group has lower scores on perceived maternal care, higher scores on perceived maternal overprotection and lower scores on emotional stability than the normal-range group. These results are similar to those reported by Páez et al. [23], who also found that poor maternal care and higher maternal overprotection during childhood are related to an insecure attachment style. Moreover, as has been mentioned above, many studies have shown the relationship between emotional stability and adult attachment styles. According to these studies, and as the current study also shows, insecure attachment involves lower levels of emotional stability [21, 22, 40].

One of the limitations of this current study is that all participants are university students. For this reason, further studies should also be made with the CAA questionnaire, but with a more heterogeneous sample that also includes workers and unemployed adults. The current study was part of a larger research that required a considerable number of questionnaires to be administered, which made it difficult to obtain samples from other environments outside the University. Within this constraint, however, we tried to make the sample more heterogeneous by including not only undergraduate students but master's degree students as well. Having said that, we fully agree in that further studies should be made with other samples to determine the extent to which the present results are replicable and generalizable.

To sum up, the current study provides objective evidence about the different profiles that can be assessed with the CAA questionnaire. It can only differentiate between two profiles: insecure attachment and normal-range. Previous studies have proposed classifications with a larger number of adult attachment styles [10], but none of them include a normal-range profile. This may be explained by the different methodological approach used in the current study, the FMA, which is more suitable than the traditional approaches used in this field. Furthermore, it should be taken into account that Stein et al. [41] have already pointed out that most adults have a style that does not quite fit into any of the prototype categories, or they have qualities in more than one prototype. In fact, although the participants in their study had the option of choosing a single attachment style from the Relationship Questionnaire (RQ) to characterize their relationships, 70% preferred to assign points to the four styles, and 28% to three styles, instead of choosing a single one. According to these authors, over the years there are more and more opportunities to generate multiple attachments, so the concordance of attachment styles across the different relationships is questionable. In fact, even 12-month-old children may show a different attachment pattern with each parent [42]. So as people develop, they may display different attachment patterns with different relationships, making it difficult to classify the attachment style of some adults within the prototypical categories. Our study, then, supports the result that a considerable percentage of adults do not have a differentiated attachment style, probably because they present some characteristics of the many different styles of attachment they have developed over the years. This might explain why there are 77.48% of adults in the current study with medium levels in the four attachment-related factors and why no more specific profiles can be identified in these participants. However, further studies should be made with other questionnaires, such as the Relationship Questionnaire (RQ) in order to determine if the number and types of profiles obtained with the FMA are the same as those proposed here. It is quite possible that further FMA studies based on other questionnaires more widely used in this field and administered to heterogeneous sample of adults

(which also includes employees, unemployed, etc., not just students) would be able to identify a higher number of attachment profiles, among them perhaps the dismissive/independent attachment style. At the same time, however, we believe that these finer distinctions would be relatively minor, so that most adults would continue to be part of the undifferentiated profile, as we found here, and only few of them will belong to the dismissive/independent or anxious/avoidant styles. In conclusion, we acknowledge that the current study is only a starting point and that further studies are needed, but we believe that this point is relevant and a clear basis. Furthermore, taking into account the results obtained here, we consider that the assessment of adult attachment styles in general should integrate both the categorical and the dimensional perspective, in order to achieve a more complete and deep understanding of these attachment styles.

## Author Contributions

**Conceptualization:** Fabia Morales-Vives.

**Data curation:** Fabia Morales-Vives, Gisela Ferré-Rey, Pere J. Ferrando.

**Formal analysis:** Fabia Morales-Vives, Pere J. Ferrando.

**Investigation:** Fabia Morales-Vives, Gisela Ferré-Rey, Pere J. Ferrando, Misericòrdia Camps.

**Methodology:** Pere J. Ferrando.

**Project administration:** Fabia Morales-Vives, Misericòrdia Camps.

**Resources:** Gisela Ferré-Rey, Misericòrdia Camps.

**Software:** Pere J. Ferrando.

**Supervision:** Fabia Morales-Vives, Misericòrdia Camps.

**Validation:** Pere J. Ferrando, Misericòrdia Camps.

**Visualization:** Fabia Morales-Vives, Misericòrdia Camps.

**Writing – original draft:** Gisela Ferré-Rey.

**Writing – review & editing:** Fabia Morales-Vives, Pere J. Ferrando.

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
