## [Decision Letter · Decision Letter 0]

7 Jun 2021

PONE-D-21-02690

Balancing typological and dimensional approaches: Assessment of adult attachment styles with Factor Mixture Analysis

PLOS ONE

Dear Dr. Morales-Vives,

Thank you for submitting your manuscript to PLOS ONE. After careful consideration, we feel that it has merit but does not fully meet PLOS ONE’s publication criteria as it currently stands. Therefore, we invite you to submit a revised version of the manuscript that addresses the points raised during the review process.

We look forward to receiving your revised manuscript.

Kind regards,

Frantisek Sudzina

Academic Editor

PLOS ONE

Journal Requirements:

3. You indicated that you had ethical approval for your study. In your Methods section, please ensure you have also stated whether you obtained consent from parents or guardians of the minors included in the study or whether the research ethics committee or IRB specifically waived the need for their consent.

5. Please upload a copy of Supporting Information Figures S1 and S2 and Tables S1-S5, which you refer to in your text on page 34.

Reviewers' comments:

Reviewer's Responses to Questions

**Comments to the Author**

1. Is the manuscript technically sound, and do the data support the conclusions?

Reviewer #1: Yes

Reviewer #2: Yes

2. Has the statistical analysis been performed appropriately and rigorously? 

Reviewer #1: Yes

Reviewer #2: Yes

3. Have the authors made all data underlying the findings in their manuscript fully available?

Reviewer #1: Yes

Reviewer #2: Yes

4. Is the manuscript presented in an intelligible fashion and written in standard English?

Reviewer #1: Yes

Reviewer #2: Yes

5. Review Comments to the Author

Reviewer #1: The introduction is clear and makes relevant points and make a reasonable case for their study. The authors discuss the Adult Attachment Scale but not the The Relationship Questionnaire – clinical version RQ–CV, which does have capacity for dimensional and categorical categories, although albeit as a fairly simple measure. There is a variant with two different ‘dismissive’ categories, which I have found useful in clinical work (Holmes B, Ruth-Lyons K. The Relationship Questionnaire – clinical version RQ–CV: introducing a profoundly distrustful attachment style. Infant Ment Health J 2006; 27: 310-325).

I note the authors cite research where they say there are problems with people wanting to elect several different RQ categories. I use the RQ-CV routinely in clinical practice and in the CV version, people can assign themselves to different styles dimensionally but then choose one category. I have used hundreds of times and not experienced problems with people wanting to choose multiple categories. I do agree with the authors that is useful to be able to do both.

The findings were consistent with their hypotheses and earlier studies. The authors note consistency with the findings of Wilhelm et al’s paper that found gender differences in impact of childhood experience on adult attachment style. This wasn’t considered in this paper, but would be worth exploring. Can this be added to the paper?

In the results, while the study was able to differentiate normal/secure and anxious/avoidant styles, which are important, it was not able to do the same for a dismissive/independent style, which is problematic as it has important interpersonal and health service utilisation implications.

Nonetheless, the discussion does make some useful points about what the various attachment types are conveying. Having ‘students only’ participants is a limitation in terms of generalisability, as noted by the authors. Perhaps the authors could speculate on what differences (if any) a wider group might make.

Style

The writing is coherent and clear. It would be possible to cut down some words as there is some repetition, e.g, text on page 18 lines 386-389 is pretty repeated on page 23 lines 505-513.

The tables are clear and well-presented.

Reviewer #2: The manuscript uses modern, sophisticated, and powerful analytical strategies and tools, and this is, in my opinion, a strength of the proposal. It is not usual to start with the strict FA approaches previous to the FMA and even less usual is to assess validity evidence using a full structural equation model. With regards to FMA in particular, I believe it is a more powerful and appropriate tool than more approximate approaches such as latent class or taxometric analysis. However, the complexity of the method also implies limitations. In particular, imposed constraints within class (measurement invariance) might impair the relative fit of models with more classes. In this respect, however, I agree with the authors’ choice and believe that these constraints are unavoidable: the more flexible and complex a model is made, the more unstable it becomes, and, even with the invariance constraints, the FMA models used in the study are very complex. Furthermore, the two-class solution fits clearly better than the competing alternatives, and the results (including those concerned with validity) are sound.

In the same line of the comments above, it is at first sight somewhat disappointing that, after so many previous multi-classes an taxometric proposals, the FMA results suggests here a quite simple, quasi-dimensional solution consisting of a general class and a relatively minor (insecure attachment) class. In principle, I believe that this parsimonious solution is essentially correct and well- supported from the data, but also believe that it has to be considered as a starting proposal that warrants further research. Thus, it is possible that the use of a psychometrically superior instrument, perhaps with a simpler factor structure, might allow to fit FMA models with less constraints, making possible a more fine-grained analysis and, perhaps, the identification of further (although possibly minor) classes. On the other hand, in order to generalize their findings, the analyses of more heterogeneous samples than the one reported in the future is needed because it may determine if the two profiles founded in this study are enough or if in this case other ones may appear.

To sum up, although this study represents a good starting point, with an appropriate methodology as well as clear and sound results for the stated objectives, it is not sufficient to provide a definitive answer on the number and kind of adult attachment profiles. Further studies are needed, using this methodology in other more well-known instruments in this field, such as the Relationship Questionnaire, using more heterogeneous samples, to know whether if the results are generalizable to other instruments and samples. The authors should discuss this more in deep in the Discussion.

6. PLOS authors have the option to publish the peer review history of their article (what does this mean?). If published, this will include your full peer review and any attached files.

Reviewer #1: No

Reviewer #2: No

---

## [Author Response · Author response to Decision Letter 0]

10 Jun 2021

Response to the Editor:

1. As the editor suggests, we have reviewed the PLOS ONE's style requirements, and we have carried out several corrections. Furthermore, we have changed the names of the files for Figure 1 and Figure 2.

2. We have reviewed our reference list to ensure that is correct and complete. In fact, we have made several changes in the references to meet the PLOS ONE's style requirements. Furthermore, we have added the new reference suggested by Reviewer 1 (doi: 10.1002/imhj.20094). We have changed one of our previous references (doi: 10.5944/ap.9.1.435) for another one (doi: 10.1016/j.rips.2016.05.002), because the doi of the previous reference does not link properly with the article. On the other hand, we have not cited any paper that have been retracted.

3. All the participants were undergraduate students older than 17. In fact, this study is focused on adult attachment styles, and for this reason one exclusion criteria to participate in this study was being under 18 years old, as we specified to the research ethics committee. We have corrected the mistake we made in the Participants’ section for which we apologize. 

4. We have uploaded our data in the ZENODO repository. The doi is https://doi.org/10.5281/zenodo.4923526

5. We have removed the section “Supporting information” on page 34, because we have not supporting tables or figures.

Response to Reviewer 1:

We are grateful to the reviewer for his/her useful recommendations. We have made several changes in the manuscript, following these comments and suggestions, as we explain below:

- As the reviewer suggests, we have described and referred the instrument Relationship Questionnaire – clinical version RQ–CV in the Introduction section. More specifically, we have explained that it includes an additional attachment-style description, labeled profoundly-distrustful attachment style. We also explain than it is a simple instrument, easy to use, and which also encompasses both categorical and dimensional perspectives. Indeed it is of potential interest for further studies.

- We also agree with the reviewer in that it is important to use both the categorical and the dimensional perspectives if a deeper understanding about adult attachment styles is to be attained. In fact, although some adults may fit primarily into one category, as the reviewer says, at the same time they may exhibit some characteristics on some other category. For this reason, the dimensional perspective complements the categorical one. We have explained this point in the revised version of the manuscript. 

- The current study is not focused on gender differences in validity relations, but it would be interesting to carry out further studies to determine whether the gender differences found by Wilhelm et al. are replicated with the instruments and the methodology of the current study. We have explained this point in the manuscript. 

- We agree with the reviewer in that having only students in our sample is a limitation, as we recognized in the manuscript. For this reason, we consider that further studies are needed with more heterogenous samples, in order to determine if more profiles can be identified, among them the dismissive/independent attachment style. However, we believe that most adults will probably belong to the undifferentiated profile, as found in the present study, and that only few of them will be in the dismissive/independent or anxious/avoidant styles.

- We have reviewed the manuscript in order to reduce some repetitions, and we have cut down some words in the lines indicated by the reviewer. We are grateful for this suggestion and the other remarks.

Response to Reviewer 2:

We are grateful for the positive and encouraging comments of the reviewer. As he/she says, FMA is a complex modeling and this involves some limitations that are unavoidable if identifiable and stable solutions are to be obtained. However, it should be taken into account that the two-class solution has clearly a better fit than the other competing solutions, and the results of both the FMA and validity analyses are consistent, as the reviewer recognizes. Therefore, we consider that the two-class solution was an appropriate choice.

The reviewer considers that, after so many previous multiple-classes and taxometric proposals, the simple FMA results we obtained here are somewhat disappointing. We understand this view and partly agree with it, but we would like to stress that our conclusions are plausible and well supported by the results found in this study, as the reviewer acknowledges. It is quite possible that more profiles could be obtained with a more heterogeneous sample, or with other instruments with a clearer factor structure. But, at the same time, we believe that this “finer” distinctions would be relatively minor and would not contradict the present results. We would like to stress again that this appears to be the first study in the field that is based on FMA, and that, compared to previous approaches, this analytical tool is considered to be more adequate for encompassing both the categorical and dimensional approaches. Overall, we consider the current study as a relevant starting point that provides solid bases results. However, further studies based on more heterogeneous samples and other instruments most widely used in this field are clearly needed in order to determine whether a larger number of profiles can be identified.

---

## [Editor Report · Decision Letter 1]

25 Jun 2021

Balancing typological and dimensional approaches: Assessment of adult attachment styles with Factor Mixture Analysis

PONE-D-21-02690R1

Dear Dr. Morales-Vives,

We’re pleased to inform you that your manuscript has been judged scientifically suitable for publication and will be formally accepted for publication once it meets all outstanding technical requirements.

Kind regards,

Frantisek Sudzina

Academic Editor

PLOS ONE
---

## [Editor Report · Acceptance letter]

29 Jun 2021

PONE-D-21-02690R1 

Balancing typological and dimensional approaches: Assessment of adult attachment styles with Factor Mixture Analysis 

Dear Dr. Morales-Vives:

I'm pleased to inform you that your manuscript has been deemed suitable for publication in PLOS ONE. Congratulations! Your manuscript is now with our production department. 

Kind regards, 

on behalf of

Dr. Frantisek Sudzina 

Academic Editor

PLOS ONE